# The Side Effects of the Most Commonly Used Group of Antibiotics in Periodontal Treatments

**DOI:** 10.3390/medsci6010006

**Published:** 2018-01-18

**Authors:** Saimir Heta, Ilma Robo

**Affiliations:** 1Pediatric Surgery, University Hospital Center of Tirana, RR. Dibra, 370, Tirana 1001 Albania; sa.heta@yahoo.com; 2Department of Periodontology, Faculty of Medical Sciences, Albanian University, Tirana 1001, Albania

**Keywords:** antibiotic, via side effects, periodontology

## Abstract

Antibiotic combinations are preferred for the treatment of periodontal diseases, with the aim of hitting the bacterial flora, according to its characteristics—aerobic, anaerobic, gram-negative, and gram-positive—with certain antibiotics that act on certain bacteria. The aim of this study is to analyze the side effects of the antibiotics used. Data on the side effects (preferably expressed in percentages) of some antibiotics, the favorites in periodontal recipes, are gathered from the literature. These data are listed according to the antibiotic used. In the case of providing a periodontal prescription, the patient is at risk of allergy (5%), nephritis (3%), hematological problems (2–2.5%), gastrointestinal problems (5.5%), disturbance in the nervous system (2%), allergic signs on the skin (5.5%), and problems with electrolytes displayed in lower percentages. Interaction with different medications is present in almost all cases. The influence on the body systems is 4% in total, the maximum value of which is expressed on the skin, and the minimum value is expressed in the nervous system. Cross allergies are at a high value because of the expressed structural similarity of antibiotics. Given a recipe, we have a balance of the percentage of side effects, the percentage of bacterial resistance, and the percentage of the success of the recommended dose of antibiotics.

## 1. Introduction

Periodontal illnesses, as a group of illnesses that are caused by a combination of bacteria and oral variable bacterial flora, require local treatment at dental clinics, and, in special cases, are also associated with oral antibiotic combinations, with systemic action. According to oral pathologies, only one antibiotic or a combination of antibiotics may be prescribed. The whole difference between these two forms of recipes exists in the dosages of antibiotics. When prescribing a single antibiotic, we prefer a dose of 500 mg, but for two antibiotic prescriptions, dosages are reduced to 250 mg for each antibiotic in the combined group of antibiotics. Antibiotic combinations are preferred to be used with the intention of hitting the bacterial flora according to its characteristics—aerobic, anaerobic, gram-negative, and gram-positive—with certain antibiotics acting on certain bacteria. Each of the selected antibiotics is effective only in the case when the total periodontal curettage is completed; otherwise, it can gradually be passed to the diagnosis of refractory periodontitis.

According to a study on the effects of combined antibiotics, amoxicillin and metronidazole are given after complete curettage; the application of antibiotics leads to much better clinical results compared to the periodontal treatment with mechanical curettage alone. According to the same study, this combination of antibiotics fights *T. forsythia* in a manner that prevents recolonization for up to 6 months after treatment. This element provides stability through the impairment expectancy of this periodontal treatment [1].

The combination of periodontal microbiology and antibiotic therapy qualifies as a normal extension of periodontal treatment, followed by a proper clinical diagnosis [2].

## 2. Side Effects of Antibiotics Prescribed in Periodontal Recipes

Data, preferably expressed in percentages, were collected from literature on the side effects of some antibiotics which are most frequently prescribed in periodontal recipes. These data are listed according to the prescribed antibiotic.

### 2.1. Penicillins

During the application, amoxicillin causes hypersensitivity, manifesting in cross reaction and sensitivity towards degradation products with alkaline hydrolysis. Allergic reactions to penicillin are as frequent as 5–8% of cases, in contrast to penicillin anaphylactic shock, which occurs in the interval of 0.05% [3].

Nephritis, eosinophilia, and hemolytic anemia are other side effects that may be associated with typical oral lesions. Nausea, vomiting, diarrhea, and gastrointestinal problems appear in the case of the application of oral doses. Vaginal candidiasis is often caused by the application of ampicillin and amoxicillin.

For penicillin, these are the efficiencies registered: anaphylactic allergic reaction, immunoglobulin E (IgE) 10%, urticaria rash; nontoxic antibiotic; eosinophilia 1–2%, 6–9%, late reaction of IgG/IgM: 1–5%; nephritis, fever, eosinophilia, hematuria: 1–2%; hematological reactions, hemolytic anemia, immune thrombocytopenia, leukopenia 1–5%; nervous system: epileptogenic action, encephalopathy; gastrointestinal system: diarrhea, pseudomembranous colitis, dysbacteriosis; cross allergy 10%; problems in the balance of electrolytes (Na, K). Broad spectrum antibiotics are not intended for epileptic women, or for those who use oral contraceptives [3].

### 2.2. Cephalosporins

Cephalosporins may cause patient hypersensibility with the same frequency as penicillins. Their chemical structure somewhat differs from penicillins, and penicillin allergic patients may not exhibit hypersensitivity to cephalosporins. Allergic reactions occur in 5–10% of cases.

For cephalosporins, the following data have been registered: non-toxic, side effects 1–10%; allergies, hives, morbilliform signs, eosinophilia: 1–2%; gastrointestinal problems 4%; 2% hematological reactions; intolerance to alcohol; nephrotoxicity, allergic interstitial nephritis. They are indicated in cases of female genital organ infections, meningitis, urinary tract infections, infections of the skin, and for surgical prophylaxis. Interaction with drugs: theophylline, streptomicin.

### 2.3. Tetracyclines

Tetracycline has gastrointestinal side effects, nausea, vomiting, and diarrhea. Tetracycline modifies the normal intestinal flora, inhibiting coliform organisms and allowing the overproduction of *Pseudomonas*, *Proteus*, and *Clostridium*. Vaginal candidiasis is associated with taking tetracycline [4]. Tetracycline is fixed to the structure of newly formed teeth, if it is taken during certain periods of pregnancy, such as at the fetal development stage. Liver toxicity occurs in cases where patients have previously had hepatic insufficiency, or when tetracycline is given intravenously. Tetracycline renal toxicity occurs when it is given together with diuretics, after nitrogen retention products. Local tissue toxicity appears with venous thrombosis, and sensitivity to light is another side effect. Nausea and dizziness occur in 35–70% percent of cases [4].

### 2.4. Macrolides

Macrolides have gastrointestinal effects, such as gastrointestinal intolerance, which is related to the stimulation of bowel motility. They may cause acute hepatitis with fever and jaundice. The state of most patients can be improved after discontinuing the drug, but the symptoms may reappear after resuming its administration.

Allergic reactions include fever, eosinophilia, and rash. Macrolides may increase serum concentrations of many drugs, such as theophylline, oral anticoagulants, cyclosporin, and metilprednizolon.

Erythromycin increases plasma concentrations of oral digoxin, increasing its bioavailability [4]. Azithromycin differs from erythromycin and from claritromycina due to its pharmacokinetic characteristics. A dose of 500 mg of azithromycin provides lower plasma concentrations of 0.4 microg/mL Azithromycin penetrates well into most tissues except the brain fluid, at concentrations 10–100 times higher than plasma concentrations. The tissue half-life is 2–4 days and ensures an elimination half-life of approximately 3 days. These properties allow azithromycin to be taken once a day and with a shorter duration of therapy. It should be taken 1 or 2 h before meals. It does not interact with other drugs, unlike erythromycin and clarithromycin [4].

Erythromycin has the following side effects: gastrointestinal problems occur in 3–4% of cases; skin allergies (1–2%); problems of the central nervous system (1–2%). It is indicated in cases of patients allergic to penicillin. It shows interaction with other medications such as theophylline.

### 2.5. Metronidazole

Antiprotozoal, as metronidazole, has a more powerful antibacterial activity against anaerobes, such as *Clostridium*. The 250 mg oral dose penetrates into the cerebrospinal fluid. Metronidazole is metabolized in the liver. Bacterial vaginosis is well treated with metronidazole. Nausea, diarrhea, stomatitis, and neutropenia are the most frequent side effects. Metronidazole offers dentists a good degree of efficiency, several advantages, and relatively minor side effects. It is the antibiotic against which suspects are still in the clinical development of resistance [2].

Based on the data provided by the literature, it appears that tetracycline causes discoloration of the teeth, gastrointestinal disorders, rare allergy, photosensitivity, nitrogen retention, and progressive uremia. It shows interactions with coumarin.

### 2.6. Clindamycin

Clindamycin has side effects like neutropenia, diarrhea, nausea, and enterocolitis. Clindamycin is displayed with the following information: skin allergy occurs in 10% of cases, while gastrointestinal disturbances occur in about 11% of cases. Indicated in the case of patients allergic to penicillin, erythromycin is the postsecondary opportunity. It is indicated for the treatment of acne and osteomyelitis. It shows interactions with theophylline. Simultaneously giving the diuretic furosemide or other antibiotics, such as vancomycin, should be avoided because they empower nephrotoxicity. Side effects are auditory damage, vertigo, ataxia and loss of balance, and others. Given alone, side effects appear as follows: it affects the gastrointestinal system, with consequences such as nausea, vomiting, and diarrhea, as well as the nervous system, causing peripheral neuropathy, encephalopathy, hallucinations, etc.; intolerance to alcohol, associated with the emergence of an unpleasant metal taste in the mouth, and the brown coloration of urine.

### 2.7. Carbapenem

Carbapenem displays the following side effects: gastrointestinal disturbances, vomiting, nausea (4%), diarrhea (3%); pseudomembranous colitis (0.16%); appearance of skin allergy, to the extent of 2.7%; nervous system disturbances occur in 3% of cases; hematological problems appear in 0.3%. Side effects are estimated at the time of the analysis of antibiotic therapy indications, with indications for the purpose for therapy or for prophylaxis. Side effects are caused because antibiotics are foreign materials in the body. They have chemical structures that can cause toxic effects. They can cause allergies through the IgE. As foreign material in the body, the biological effects are also expressed in the body.

## 3. Discussion

When granting antibiotics, it is best to rely on microbiology diagnostics. This means that the sensitivity of certain bacteria to antibiotics must be found. Thus, we are convinced that the antibiotic effect will be adequate. Plaque and biofilm must be mechanically removed before giving antibiotics [5]. If antibiotics are not selected properly, higher pathogenicity is allowed through the transfer of genetic material for increased virulence and antibiotic resistance in oral microflora [6].

In cases of periodontal treatment, this type of control is difficult because the laboratory conditions for planting and microbiology diagnostics of oral bacterial flora are incomplete. The prescription of antibiotics is carried out on the basis data obtained from the periodontal clinical examination of the patient. Clinical examination reveals the presence of certain bacteria, such as plaque color change. It is known that the typical color of plaque changes depending on what bacteria or bacterial compound are included in the plaque structure. The color ranges from white to yellow, orange, green, cherry, or coffee. These color fluctuations show the presence of bacteria expressing the specific layer that gives color. Knowing the characteristics of bacteria and comparing the sensitivity, doctors can also prescribe a combination of antibiotics. We strive for a broad-spectrum antibiotic to combine with narrow-spectrum antibiotics, and, further, for a field with the element fighting both aerobes and anaerobes. However, after an oral antibiotic is taken, its absorption causes a reaction in the system throughout the organs; that is the road that takes this hematological antibiotic to the gingiva and other organs. Doses of certain antibiotics express their concentration in area of gums. Besides the positive effect of the antibiotic, side effects may also be encountered expressed in organs and other systems of the body.

Based on the above data about the side effects of antibiotics applied in the treatment of periodontal diseases, the obtained results show that patients are at the risk of:-Allergy (5%),-Nephritis (3%),-Hematological problems (2–2.5%),-Gastrointestinal problems (5.5%),-Disturbance in the nervous system (2%),-Signs of allergy on the skin (5.5%),-Problems with electrolytes displayed in lower percentages.

Allergies against drugs are expressed as main and primary complications in 5% of cases. The hematologic system, gastrointestinal system, nervous system, cardiovascular system, and kidneys all appear in a list of issues that manifest as side effects. Interaction with different medications is present.

The influence on the body systems is 4% in total, the maximum value of which is expressed on the skin, and the minimum value of which is expressed in the nervous system. Jaundice is a side effect that occurs as result allergic reactions in other organs, with the added value of IgG-IgM complexes (non-IgE). Most antibiotics are associated with disorders in the normal function of platelets and red blood cells. They affect intestinal flora, necessary for the absorption of vitamin K, and can lead to bleeding problems.

In this sense, the percentage of gastrointestinal disorders (5.5%) affects the percentage of hematological problems (2.5%). It should be noted that those impacts are proportional. Taking antibiotics affects the effects of other drugs that patients take, in the context of binding proteins for transport. We stress that the nervous system disturbances are present in lower percentages and only in the cases of overdose of antibiotics. With the termination of treatment with the antibiotic, everything returns to normal.

Side effects are divided by the class and proximity of antibiotics with each other; this is linked to the similarity in their chemical structure. The percentage of bacterial resistance to the antibiotic classes is not reflected in the results. This is because the purpose of the study was a simple analysis of values of percentages of side effects only, and not the effects on bacterial strains. So, when giving a prescription, we balance between the percentage of side effects, the percentage of bacterial resistance, and the percentage of success based on the recommended dose of antibiotics. Local drug delivery as an active treatment or maintenance therapy depends on clinical findings, responses to treatment described in the literature, desired clinical outcomes, and patients’ dental and medical histories, including their past usage of antimicrobials [7]. A meta-analytic study demonstrated that statistically significant—though not clinically substantial—improvement could be achieved in cases of chronic periodontitis when local delivery of tetracycline was used as an adjunct to scaling and rootplaning [8]. Tetracycline fiber therapy along with scaling and rootplaning improves the healing outcome, namely, the reduction in pocket depth and gain in clinical attachment level, when compared to scaling and rootplaning alone [9]. Studies addressing metronidazole utilization in a variety of clinical conditions demonstrate that its routine use does not enhance rootplaning [10]. We conclude that systemic metronidazole given 250 mg for 7 days in conjunction with debridement of the tooth surfaces can significantly reduce the need for periodontal surgery compared to the standard regimen which included only debridement [11]. These latest data, based on published studies, point out once again that any treatment with antibiotics, systemic or local, only reaches its maximum periodontal healing effect, if applied after the removal of bacterial plaque from the affected teeth. The healing effects at periodontal structures are balanced by the side effects of antibiotic therapy.

## 4. Conclusions

Among the drugs researched, tetracycline and metronidazole were the only drugs without accurate data in percentages about their side effects. Cross allergies appear at high frequency because of the structural similarity of these antibiotics. The percentage of bacterial resistance to the antibiotic classes is not reflected because the purpose of the study was a simple analysis of percentage values of the side effects only and not of the effects on bacterial strains.

So, in giving a prescription, we balance the percentage of side effects, the percentage of bacterial resistance, and the percentage of therapy success, based on the recommended dose of antibiotics.

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
