# Peer review of "The Side Effects of the Most Commonly Used Group of Antibiotics in Periodontal Treatments"

_medsci, 2018, doi:10.3390/medsci6010006_

Reviewer 1 Report

This manuscript complied the side-effects of various antibiotics used in the treatment of periodontitis. The introduction should include the various bacteria that cause periodontitis and should classify the bacteria (Gram-positive, negative/ aerobic, anaerobic) and the major etiological agent be defined. The authors talk about the choice of antibiotic being dependent on the type of bacteria present, yet there is no mention in the introduction about what the current antibiotic regimen based on the bacterial type. Some type of chart type representation should be used to ensure readability of the data.

Author Response

After carefully  reading the opinions expressed, we think that this is not the purpose of  our topic. We do not intend to discuss the type of bacteria that causes the periodontal diseases, and that  antibiotic is acting on, or no. But, we have to say that  these are the antibiotics that are given in prescriptions in the cases of  periodontal diseases and they have their side effects, how much the patient  is at risk to these effects.
It is not for the  purpose of the article, to give the prescriptions that the doctor writes  when he locates a certain periodontal diagnosis.

Reviewer 2 Report

This short review by Heta and Robo presents the side effects of a group of antibiotics that are used in periodontal treatments. My comments are listed below:

·         Although it is a review the number of references used is limited, only six references while the most recently one was published in 2012. A more extended survey should be performed to enrich the review.  

·         All bacteria should be written in italics, e.g. T. forsythia

·         There were several repetitions throughout the text that should be avoided

·         Authors should check their text carefully for typing errors such as “war-binding proteins, lane 172” that don’t make sense.

Author Response

In the article it was noted that there are no figures per cent for the level of side effects of the mentioned antibiotics. It  is not the year 2012, the last year from which the data came from, but  only those references had figures in percentages of the side effects of  antibiotics.
Other requirements will fill in the in the article.